# Whole-Genome Resequencing Reveals Selection Signatures of Abigar Cattle for Local Adaptation

**DOI:** 10.3390/ani13203269

**Published:** 2023-10-19

**Authors:** Wondossen Ayalew, Xiaoyun Wu, Getinet Mekuriaw Tarekegn, Tesfaye Sisay Tessema, Rakan Naboulsi, Renaud Van Damme, Erik Bongcam-Rudloff, Zewdu Edea, Solomon Enquahone, Ping Yan

**Affiliations:** 1Key Laboratory of Animal Genetics and Breeding on Tibetan Plateau, Ministry of Agriculture and Rural Affairs, Key Laboratory of Yak Breeding Engineering, Lanzhou Institute of Husbandry and Pharmaceutical Sciences, Chinese Academy of Agricultural Sciences, Lanzhou 730050, China; wondessenayalew9@gmail.com; 2Institute of Biotechnology, Addis Ababa University, Addis Ababa P.O. Box 1176, Ethiopia; getinet.tarekegn@sruc.ac.uk (G.M.T.); tesfu74@yahoo.com (T.S.T.);; 3Scotland’s Rural College (SRUC), Roslin Institute Building, University of Edinburgh, Edinburgh EH25 9RG, UK; 4Childhood Cancer Research Unit, Department of Women’s and Children’s Health, Karolinska Institute, Tomtebodavägen 18A, 17177 Stockholm, Sweden; 5Department of Animal Breeding and Genetics, Bioinformatics Section, Swedish University of Agricultural Sciences, P.O. Box 7023, S-750 07 Uppsala, Sweden; renaud.van.damme@slu.se (R.V.D.); erik.bongcam@slu.se (E.B.-R.); 6Ethiopian Bio and Emerging Technology Institute, Addis Ababa P.O. Box 5954, Ethiopia; zededeaget@gmail.com

**Keywords:** Abigar cattle, adaptation, selection signature, thermotolerance, tropical environment

## Abstract

**Simple Summary:**

Abigar cattle, native to southwestern Ethiopia’s hot and humid environment, are recognized for their adaptability and vital contribution to local livelihoods and the livestock value chain. Investigating their genetic basis for adaptive traits is crucial for sustainable use. However, there is a paucity of studies on genomic diversity, population structure, and selection signatures of Abigar cattle. This study introduces the first whole-genome sequencing of Abigar cattle, revealing genes linked to heat tolerance, immune response, and stress resilience in tropical conditions. These findings offer essential genomic insights for future Abigar cattle breeding.

**Abstract:**

Over time, indigenous cattle breeds have developed disease resistance, heat tolerance, and adaptability to harsh environments. Deciphering the genetic mechanisms underlying adaptive traits is crucial for their improvement and sustainable utilization. For the first time, we performed whole-genome sequencing to unveil the genomic diversity, population structure, and selection signatures of Abigar cattle living in a tropical environment. The population structure analysis revealed that Abigar cattle exhibit high nucleotide diversity and heterozygosity, with low runs of homozygosity and linkage disequilibrium, suggesting a genetic landscape less constrained by inbreeding and enriched by diversity. Using nucleotide diversity (Pi) and population differentiation (*F_ST_*) selection scan methods, we identified 83 shared genes that are likely associated with tropical adaption. The functional annotation analysis revealed that some of these genes are potentially linked to heat tolerance (*HOXC13*, *DNAJC18*, and *RXFP2*), immune response (*IRAK3*, *MZB1*, and *STING1*), and oxidative stress response (*SLC23A1*). Given the wider spreading impacts of climate change on cattle production, understanding the genetic mechanisms of adaptation of local breeds becomes crucial to better respond to climate and environmental changes. In this context, our finding establishes a foundation for further research into the mechanisms underpinning cattle adaptation to tropical environments.

## 1. Introduction

African cattle have been playing an indispensable role in shaping the development and interdependence of societies, economies, and ecosystems across the continent for generations. As a fundamental pillar of African agriculture, these magnificent animals hold immense significance in the lives of millions, contributing to livelihoods, food security, cultural heritage, and economic growth [1,2]. African cattle have undergone a remarkable journey of evolution, shaped by centuries of natural selection and adaptation to diverse ecosystems and climates. Spanning the arid landscapes of the Sahelian regions to the fertile grasslands of the Savannah, these cattle populations have successfully adapted to scarcity of feed and water resources, extreme temperature fluctuations, and region-specific diseases and parasites [3,4,5,6]. Their ability to thrive in such challenging environments stands as a testament to their resilience, enabling them to overcome environmental constraints. Moreover, the high within and between-breed genetic diversity further enhances their adaptability, ensuring their continued provision of invaluable resources to the communities they serve. Thus, the detection of specific genomic regions influenced by both human selection and environmental selective pressure is essential for understanding how changes in the genome contribute to variations in phenotype and adaptation to extreme environments. This information is key to enhancing animal breeding techniques, leading to improvements in production, health, and overall welfare outcomes [7]. Although the content is rich in cattle genetic resources, limited studies have been conducted on the adaptive attributes of African cattle breeds [5,6,8,9]. This highlights the need for further research into the distinctive genetic factors that underlie adaptive traits within unexplored individual cattle breeds [10,11].

In the continent, Ethiopia has broad and diverse bovine genetic resources, with over 28 indigenous cattle breeds inhabiting the diverse geographic and climate conditions [12]. These indigenous cattle breeds have long been serving as a major labor force in agricultural production and providing milk, meat, and other byproducts with the intrinsic worth of parasite resistance, utilization of roughage-based diets, and tolerance to extreme climates [6]. Ethiopian indigenous cattle are broadly classified into four groups: Zebu, Zenga, Sanga, and taurine [3]. Among indigenous cattle breeds, the Abigar cattle, classified as “Sanga”, primarily inhabit the border region between Ethiopia and South Sudan [12]. They have a significant presence in Ethiopia, particularly in the Akobo area of the Gambella region. The Sanga cattle are known for their adaptability to local environments characterized by high temperatures and high burdens of disease and parasites and are historically important for the livelihoods of communities in the region [13]. Indeed, the Abigar breed possesses remarkable adaptive traits that enable them to thrive in challenging environments. Over generations, these adaptive characteristics have evolved, allowing the cattle to cope effectively with harsh conditions, including frequent disease outbreaks, drought, seasonal feed and water shortages, and high temperature and heat loads [14]. The unique combination of these adaptive traits makes the Abigar cattle well-suited for their specific ecological niche and plays a vital role in the livelihoods of local communities, providing them with valuable resources like milk, meat, and draught power while contributing to the preservation of cultural heritage. To our knowledge, there is a dearth of genomic information about Abigar cattle. To advance our insight into the unique genomic architecture of the breed, we performed whole-genome sequencing. This endeavor unveiled an extensive list of candidate genes linked to immune response and thermotolerance functions contributing to the breed’s adaptive characteristics for further improvement and sustainable utilization.

## 2. Materials and Methods

### 2.1. Sequence Data

Whole-blood samples (10 mL) were collected from 10 Abigar and 10 Barka (Begait) cattle breeds in their natural breeding tracts. DNA extraction was carried out at the Institute of Biotechnology, Addis Ababa University, Ethiopia, using a Tiangen genomic DNA extraction kit following the manufacturer’s protocols (https://en.tiangen.com/content/details_43_4224.html, accessed on 8 June 2022). The paired-end libraries with an average insert size of 500 bp were constructed for each individual, with an average read length of 150 bp. The whole-genome sequencing (WGS) was performed using the MGI-SEQ 2000 platform by Frasergen Bioinformatics Co., Ltd. (Wuhan, China). To investigate the genetic relationship between Ethiopian cattle and other cattle breeds, we retrieved additional WGS samples of African and European taurine and African Sanga cattle breeds from the NCBI database (Appendix A).

### 2.2. Quality Control, Read Mapping and Variant Calling

To confirm the quality of the raw sequencing data, we performed a per-base sequence quality check using the fastQC v0.11.8 software (https://www.bioinformatics.babraham.ac.uk/projects/fastqc/, accessed on 3 December 2022). Then, the raw sequences were subjected to quality control and trimmed using the Trimomatic v0.39 [15] using default parameters. This step ensured the removal of adapter sequences, low-quality reads, and bases with low-quality scores. The pair-end sequence reads were mapped against the latest *Bos taurus* reference genome (ARS-UCD1.2) [16] using BWA-MEM (v0.7.13-r1126) with default settings [17]. SAMtools version 1.9 was employed to convert SAM files to the BAM format and subsequently sort the resulting BAM files by contigs [18]. To account for duplicate reads arising from PCR amplification artifacts, the MarkDuplicates function of Picard tools v2.27.4 was used (https://broadinstitute.github.io/picard/, accessed on 31 August 2023). The identification of short variants (SNPs) was performed following the GATK 4.3.0 best practice protocol (https://gatk.broadinstitute.org/hc/en-us/articles/360035535932-Germline-short-variant%20discovery, accessed on 22 June 2023). The protocol involved utilizing the HaplotypeCaller function on individual samples to detect variants, followed by the consolidation of individual samples and joint genotyping using the GenotypeGVCFs function. Furthermore, a two-step machine learning model called variant quality score recalibration (VQSR) was used to enhance the quality and reliability of variant calls. A set of high-quality variants that underwent careful curation and validation were utilized to train the VQSR model. After filtering against poor-quality sequence and non-variant data, a total of 33,522,977 biallelic, autosomal SNPs were retained for downstream analysis. Furthermore, the extent of missing genetic data within the individual sample and the interrelationship among the other samples were evaluated using “--missing-indv” and “--relatedness”, respectively, provided by the vcftools v0.1.15 [19]. Fortunately, we did not remove any sample due to excessive missing data (>20%) and high relatedness (>0.8) among individuals belonging to different breeds [8]. Finally, for each cattle breed, the total numbers of SNPs and the transition to transversion (Ts/Tv) ratio were estimated using bcftools ver 1.8 [20].

### 2.3. Genomic Diversity and Population Structure

After variant calling and obtaining the variant call set, we performed genomic diversity analysis to explore the patterns of genetic variation within and between populations. The vcftools v0.1.15 [19] was employed to estimate the average nucleotide diversity (π) and population genetic differentiation (*F_ST_*) within 100 kb windows, with a step size of 50 kb, along the bovine autosomes. The observed heterozygosity was determined as the fraction of total heterozygous SNPs to the total number of sites considered within each genome, using the ARS-UCD1.2 taurine cattle reference genome as the basis for analysis. The observed and expected heterozygosity of each population were computed by employing the “--het” option in PLINK v1.9 [21]. Subsequently, the calculated values were averaged for each population to obtain representative measures.

Principal component analysis (PCA) and admixture analysis were employed to infer population structure and admixture levels. These analyses were started with a set of high-quality autosomal SNPs, and then SNPs with minor allele frequency (MAF) less than 0.05 were filtered out, resulting in 23,963,154 SNPs. We further pruned the filtered SNPs for high levels of pairwise linkage disequilibrium (LD) using Plink 1.9 [21] with the parameter (--indep-pairwise 50 10 0.2) and removed SNPs with more than 10% missing genotypes (--geno 0.1) using vcftools [19]. Subsequently, we used pruned SNPs (1,314,830) and employed the block relaxation algorithm in ADMIXTURE ver 1.3.0 software [22] to infer the admixture levels of the study populations with K values ranging from 2 to 10. The optimal K value was obtained according to the cross-validation (CV) value (Appendix A). The admixture plot was visualized using the R package. In addition, we used the same dataset to generate eigenvalues PC with PLINK 1.9 and visualized with ggplo2 in the R ver 4.3 environments [23]. Furthermore, the neighbor-joining tree was constructed based on pairwise genetic distances using splitsTree v4.19.1 [24].

### 2.4. Linkage Disequilibrium and Run of Homozygosity

The decay of linkage disequilibrium (LD) with the physical distance between SNPs was calculated and displayed using the PopLDdecay software ver. 3.41 [25] with its default settings. The PLINK 1.9 software was used in a sliding window of 50 SNPs to identify runs of homozygosity (ROHs). The following settings were used to define ROH: (1) a minimum required density of 50; (2) allowing a maximum of 3 heterozygotes in a window; (3) permitting a maximum of 5 missing calls in a window. The number and length of ROHs were estimated for each breed, and the length of ROHs was categorized into three groups: 0.5–1 Mb, 1–2 Mb, and >2 Mb [26].

### 2.5. Genome-Wide Selection Sweeps

To elucidate the positive signatures of selection in Abigar cattle, a within-population genomic scan method (the nucleotide diversity; Pi) was employed. Additionally, the population differentiation between the Abigar and Holstein cattle breeds was estimated by Weir and Cockerham statistics (*F_ST_*) [27]. To implement these methods, we adopted a sliding window approach with window sizes set at 100 kb and a step size of 50 kb using VCFtools [19]. The top significant 0.05% genomic regions of each selection scan method was selected, and the adjacent windows were merged into a single region. These regions were subsequently annotated using the Ensembl Biomart online annotation tool (http://useast.ensembl.org/index.html, accessed on 1 August 2023), employing the ARS-UCD1.2 bovine reference genome [16]. The Database for Annotation Visualization and Integrated Discovery (DAVID, https://david.ncifcrf.gov/content.jsp?file=release.html, accessed on 1 August 2023) was employed to unravel the gene ontology (GO), pathways, and enriched terms of the candidate genes [28].

## 3. Results

### 3.1. Sequence Reads and Variant Statistics

To better understand genomic variations in Ethiopian indigenous cattle populations, we performed whole-genome sequencing of ten Abigar (ABI) and ten Barca (BAR) cattle and compared them with publicly available genomic data from Ankole (ANK), NDama (NDA), and Holstein (HOL) cattle breeds (Appendix A). The clean sequencing reads were mapped to the Bos taurus reference genome (ARS-UCD1.2) using the BWA MEM algorithm, resulting in a 99% alignment rate. As expected, the number of SNPs in taurine cattle was significantly lower when compared to African Bos indicus (Appendix A). Notably, the Abigar breed exhibited the highest number of variants (27,155,787), whereas Holstein cattle showed the lowest SNP number (10,331,162). This difference in SNP numbers between Bos indicus and Bos taurus cattle breeds aligns with previous research findings [5,8]. The average transition versus transversion (Ts/Tv) ratio for the five cattle breeds was found to be 2.31 (Appendix A). A higher ratio indicates a reduced likelihood of false-positive variants, which in turn enhances the reliability of the genetic data used for the subsequent analyses.

### 3.2. Population Genetic Structure

We performed Principal Component (PCA), admixture, and neighbor-joining (NJ) analyses to explore the genetic relationships among Ethiopian and reference cattle breeds, including African Sanga (Ankole), African taurine (NDama), and the commercial cattle breed (Holstein) (Appendix A). The PCA results exhibited distinct separation among the studied breeds according to their geographic origins (Figure 1a). PC1 and PC2 accounted for 10.44% and 3.95% of the total variation, respectively, and clearly separated the individuals into taurine and indicine breeds, with the Ankole cattle at an intermediate position. In the admixture analysis, when K = 2, the cattle breeds exhibited genetic differentiation into Bos taurus and Bos indicus ancestry. At K = 4, the admixture plot captured the highest observed ancestry proportion, which was further substantiated by the lowest coefficient of error variance (Appendix A). The Abigar breed, despite its classification among other African Sanga breeds, exhibits a substantial Bos indicus genetic background in contrast to the Ankole cattle. The Neighbor-Joining (NJ) tree analysis further supported the results obtained based on PC and admixture analyses. The NJ tree displayed a clear separation of the Sanga and zebu cattle breeds from African and European Bos taurus breeds. The admixture result revealed a substantial genomic share of the Abigar with the Ankole cattle breed, reflecting their common ancestry or historical genetic interactions (Figure 1c). 

### 3.3. Patterns of Genomic Diversity

Figure 2 illustrates the average nucleotide diversities observed among the studied breeds. Notably, the Abigar and Barca breeds demonstrated the highest nucleotide diversities, whereas the Holstein breed exhibited the lowest nucleotide diversity, consistent with the previous findings [5,8]. Similarly, higher heterozygosity was observed in Abigar and Barca breeds, while the lowest heterozygosity was detected in the Ankole breed (Figure 2e). The population differentiation (*F_ST_*) between African zebu and taurine breeds revealed significant genetic divergences. Conversely, moderate differentiation was found between the African Sanga and zebu breeds (Table 1). These moderate population differentiations among African cattle breeds provide insights into their common genetic backgrounds and possible gene flow. To investigate the genomic landscape and genetic history of populations, we categorized runs of homozygosity (ROH) into three groups based on their length: 0.5–1 Mb, 1–2 Mb, and >2 Mb (Figure 2c). As expected, the Holstein breed showed a significant accumulation of ROH in all ROH classes (Figure 2c). The observed high LD in short distance and the long ROH in the Holstein cattle breed suggests potential inbreeding, which could be attributed to long-term artificial selection practices (Figure 2d).

### 3.4. Genome-Wide Selective Sweeps

The nucleotide diversity (Pi) and population differentiation (*F_ST_*) selection scan methods were employed to uncover selection sweeps associated with adaptive traits in Abigar cattle. In both methodologies, we designated the top 0.5% of genomic regions as candidates under selection. A total of 310 and 323 genes were detected by the Pi and *F_ST_* methods, respectively (Appendix A). Of these, 83 genes were detected by both selection scan methods (Figure 3c, Appendix A). Interestingly, a substantial number of the shared genes with potential roles for tropical environment adaptations have been previously identified in different African zebus (Table 2). The identification of adaptation-related genes using both selection scan methods, aligned with prior research, underscores the Abigar cattle breed’s harbored adaptive attributes for thriving in tropical environments. Notably strong signals of differentiation observed in the regions harboring well-known candidate genes were associated with immune response genes (*WIF1*, *MZB1*, *SIRT1*, *STING1*, *IRAK3*, and *HSPA9*), oxidative stress response (*SLC23A1*), and heat tolerance genes (*ASIP*, *DNAJC18*, *HOXC13*, *HSF4*, and *RXFP2*) (Table 2; Appendix A). The genome-wide distribution of Pi and *F_ST_* values is presented in Figure 3a,b.

The functional annotation of gene sets enrichment analysis revealed several statistically significant (*p* ≤ 0.05, Bonferroni correction) enriched biological processes (Appendix A). Here, we focused on genes and GO terms related to tropical environment adaptation. Among the significant GO terms, those related to tropical environment adaptations include anterior/posterior pattern specification (GO:0009952), skeletal system development (GO:0001501), positive regulation of immune response (GO:0050778), cellular response to insulin stimulus (GO:0032869), and cellular response to stress (GO:0033554) (Appendix A).

## 4. Discussion

African cattle breeds exhibit an extensive distribution across the continent, ranging from the Afro-alpine to the Afar depression. Unraveling the genetic diversity, population structure, and specific selection sweeps of these cattle breeds provides a valuable asset for genetic improvements, sustainable agriculture, and conservation initiatives. The PCs and admixture analysis revealed clear genetic differentiations between Abigar cattle and *Bos taurus* breeds (Figure 1a,c). The distinct separation of Abigar and taurine cattle breeds provides crucial insights into the genetic structure and ancestry of the cattle populations, highlighting their unique evolutionary paths and genetic backgrounds, which have been shaped by historical selection, environmental pressures, and breeding practices [5,36].

In our study, the Abigar and Barca cattle breeds showed a relatively higher nucleotide diversity. The higher nucleotide diversity in these cattle breeds could be attributed to weak artificial selection histories compared to the Holstein cattle breed. This aligns with Kim et al. [5] and Terefe et al. [9], indicating that indigenous Ethiopian cattle breeds tend to preserve greater genetic variation, likely due to the country being a gateway from the center of domestication to the African continent and experiencing less intensive selective breeding practices. Likewise, the estimated observed heterozygosity in Ethiopian cattle revealed the high level of diversity within the studied populations. Notably, the Abigar and Barca cattle breeds exhibit higher genetic variation, which can be plausibly attributed to the absence of substantial selection pressure owing to the lack of effective breeding programs. Our finding is consistent with the reported estimates for indigenous cattle breeds of Ethiopia [9] and Sudanese zebu [8].

Runs of Homozygosity (ROH) are defined as contiguous stretches of the genome where an individual inherits identical alleles from both parents [37]. In our study, a significant proportion of ROH, identified across all breeds, was observed to fall within the length range of 0.5–1 Mb (Figure 2c). Notably, the Abigar breed exhibited the lowest ROH count in all ROH length categories, indicating a comparatively higher level of genetic diversity. The high genetic diversity suggests a potential adaptive advantage for the Abigar cattle breed, enabling them to better cope with various environmental stressors. Conversely, the Holstein breed exhibited substantial linkage disequilibrium (LD) in short genomic distances (Figure 2d) and a notable prevalence of long ROH, which may indicate the presence of potential inbreeding. These observed characteristics in Holstein breeds are likely attributed to long-term artificial selection practices [5,36].

Indigenous African cattle breeds’ exhibit unique resilience and resistance to a range of challenging environmental pressures compared to their commercial counterparts. This resilience not only highlights the adaptive capabilities of these breeds but also emphasizes the potential value they hold for sustainable livestock management and breeding programs, especially in the face of changing and unpredictable environmental circumstances. In this analysis, we have identified several candidate genes that shed light on the adaptive significance of Abigar cattle in hot and humid tropical environments marked by multiple environmental stressors, disease prevalence, limited feed resources, and parasitic challenges. Interestingly, the genes found within the identified candidate regions in our current investigation exhibit functions closely tied to traits encompassing heat tolerance, immune response, and the ability to counteract oxidative stress (Table 2 and Appendix A).

Heat stress is one of the challenges that significantly shapes the adaptive landscape of cattle populations. The constellation of candidate genes, notably *ASIP*, *DNAJC18, HOXC13*, *HOXC12*, *HSF4*, and *RXFP2* (Table 2 and Appendix A), holds profound implications for thermal adaptations. The bovine coat acts as a vital barrier against solar radiation and environmental stressors. Hair coat length, skin pigmentation, and coat color are often proposed to influence heat tolerance [38,39]. Dark-coated animals absorb more heat from solar radiation than their light-coated counterparts [40]. Cattle adapted to arid regions feature efficient heat dissipation through traits like smooth, short, and thin hair, attributed to the slick hair gene [41].

Interestingly, Abigar cattle are characterized by their predominant coat colors of white and grey [12,14]. The prevalence of white coloration in Abigar cattle is likely attributed to the involvement of the agouti signaling protein (*ASIP*), a key player in pigmentation. *ASIP* reduces eumelanin production by downregulating the melanocortin 1 receptor (*MC1R*) while simultaneously promoting pheomelanin production [42]. Beyond its genetic underpinnings, the white coat color in Abigar cattle holds substantial adaptive significance in their natural habitat. White coats possess the unique ability to reflect sunlight and mitigate heat absorption, effectively counteracting the impact of solar radiation. This attribute provides a unique advantage in the hot environmental conditions where Abigar cattle thrive, offering mechanisms for heat adaptation. Studies have indicated that an increased copy number of the *ASIP* gene might account for white pigmentation in goats [43] and sheep [44]. Additionally, in Nellore cattle, which have been selectively bred for a white coat, reduced *ASIP* expression led to elevated eumelanin production and, subsequently, a darker coat color [45]. Similarly, a loss-of-function mutation in the *ASIP* gene’s coding region frequently results in a recessive black coat color in rabbits [46].

We have also identified the *RXFP2* gene (BTA12: 29.21–29.27 Mb), previously reported for its pleiotropic effects encompassing both reproductive and horn development functions [31,47,48]. The Abigar cattle’s horns exhibit a distinctive elongated structure that extends upwards, resembling an oval configuration. However, these morphological traits extend far beyond mere aesthetics; they are intricately linked with adaptation. The distinctive structure of tropical cattle horns is believed to serve as a thermal adaptation. The significant surface area of the vascular bed, combined with the thin keratin sheath, facilitates rapid heat dissipation following strenuous activities. Furthermore, the vertical orientation of these horns on tropical bovids optimally places them in areas of heightened airflow. This arrangement effectively enhances the rapid dispersion of heat, particularly during intense, high-speed evasive maneuvers used to evade predators [49]. Similarly, Ben-Jemaa et al. [29] have highlighted that horns potentially play a role in the thermoregulation of Creole cattle. Given that the horn’s core is connected to the sinus, it is plausible that the horns contribute to nasal heat exchange. This mechanism significantly minimizes water loss by cooling the exhaled air, resulting in water condensation and reabsorption [50].

The homeobox genes (e.g., *HOXC12* and *HOXC13*) are integral developmental regulators intricately involved in shaping essential morphological traits that actively drive hair follicle differentiation, growth, and overall development [51]. Their influence on the regulation of genes associated with keratin differentiation holds profound significance, contributing significantly to the adaptive responses to heat observed in both cattle and goats [6,34]. Specifically, the *HOXC13* gene is a key determinant impacting skin thickness. The combined factors of skin thickness and the abundance of hair follicles play a pivotal role in finely tuned body temperature regulation. For instance, animals with thicker skin, such as the heat-tolerant *Bos indicus* breeds of cattle, exhibit improved thermoregulation compared to their heat-sensitive counterparts like the *Bos taurus* breeds [52]. Moreover, the role of heat shock factor 4 (HSF4) is particularly significant within the family of heat shock transcription factors. These regulatory proteins hold the reins in orchestrating cellular responses to a spectrum of stressors, with a primary focus on heat-induced stress [32]. The dynamic involvement of these transcription factors in modulating heat shock proteins during episodes of thermal stress has been documented across various African cattle populations [6,53,54]. Aside from the genes mentioned above, the *DNAJC18* gene encodes for heat shock proteins that play vital roles in facilitating protein folding, directing misfolded proteins towards degradation pathways, and upholding protein equilibrium within the cellular environment. The positive selection signals around this region are further confirmed by significantly lower Pi and high *F_ST_* values (Figure 4). This gene has been previously identified and associated with safeguarding cellular integrity across a diverse spectrum of stress scenarios, prominently including instances of heat stress [8,9,35].

For generations, the African continent has faced multitudes of selection pressures, including diseases and parasites. This prolonged exposure has nurtured potent innate and acquired immune responses, empowering the ability to combat an extensive range of diseases and parasitic challenges. For instance, we identified a highly significant genomic region on BTA7: 50.64–50.74 Mbp encompassing three immune response genes (*MZB1*, *SLC23A1*, and *STING1*), exhibiting a robust positive selection signal in Abigar cattle (Table 2). *MZB1* encodes a protein vital for the immune response and B cell function, playing a pivotal role in the adaptive immune system by generating antibodies that target specific pathogens [55,56]. *SLC23A1* pertains to sodium-dependent membrane transporters, crucial for human vitamin C metabolism. It regulates dietary intake, reabsorption, and tissue distribution of vitamin C [57]. Furthermore, hot environments intensify heat stress and physical activity, elevating oxidative stress. This increases the demand for antioxidants like vitamin C, which responds to harmful free radicals generated during cellular processes, mitigating cellular damage and inflammation. Maintaining adequate vitamin C levels proves essential for the robust production and functioning of immune cells, including key players like neutrophils, lymphocytes, and phagocytes, all integral for effective immune responses [58]. Additionally, *STING* emerges as a transmembrane protein situated on the endoplasmic reticulum. Its pivotal function lies in the innate immune signaling response, fortifying the host against viral and bacterial infections [59]. On chromosome 5, we identified anti-inflammatory genes, namely *GRIP1* and *IRAK3* (Table 2). In mice, *GRIP1* plays a pivotal role in augmenting the anti-inflammatory effects of glucocorticoids, and its deficiency results in heightened sensitivity to inflammatory challenges [60]. On the other hand, *IRAK3* is a component of the Toll-like receptor (TLR) signaling pathway, tasked with the regulation of immune responses. Specifically, *IRAK3* operates as a negative regulator of TLR signaling, adjusting the innate host defense mechanisms to modulate the extent and duration of excessive inflammation [61,62].

Last but not least, the regions inhabited by Abigar cattle are marked by food and water scarcities. We further elucidated previously identified genes on chromosome 16 (*FAAP20* and *SKI*) (Table 2), primarily associated with adaptive metabolic strategies involving insulin signaling, glucose homeostasis, and fat metabolism [8]. In general, these African cattle-specific selective sweeps are evidence of shared historical selection footprints and introgression, most likely due to their ancestral, geographical, and husbandry system acquaintances for resilience to the tropical selection pressures. This may also reflect the pleiotropic effects of genes on other relevant adaptive and agro–economic traits. However, these results would need further fine-mapping and functional genomic studies.

## 5. Conclusions

Indigenous African cattle breeds exhibit pronounced resilience and resistance to environmental pressures, holding immense potential for sustainable livestock management. This study unveiled genes that have undergone positive selection in Abigar cattle. These genes contribute to various biological and cellular functions, collectively shaping the adaptive characteristics of this cattle breed within tropical environments. The genes are mainly associated with heat tolerance, immune response, and oxidative stress counteraction and have previously been elucidated in other African cattle breeds. The shared African cattle-specific adaptation genes underscore their potential value for future breeding programs and sustainable livestock management in the face of climate change and breeding objectives.

## Figures and Tables

**Figure 1 animals-13-03269-f001:**
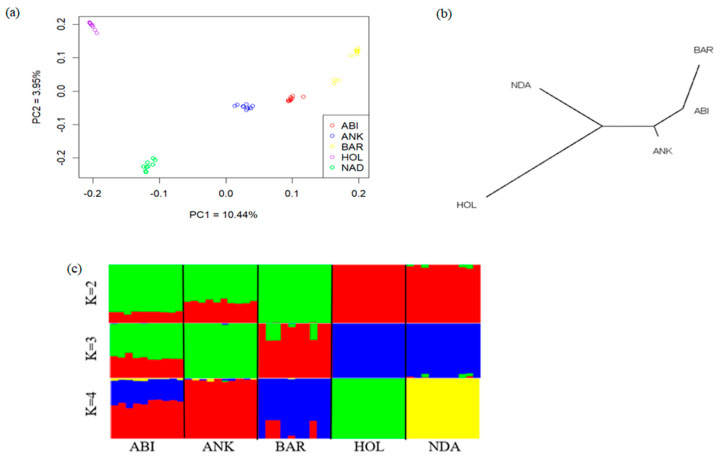
Genetic diversity and population structure of the five studied breeds (ABI = Abigar, ANK = Ankole, BAR = Barca, HOL = Holstein, NDA = N’Dama). (**a**) Principal component plots for the first two PCAs, (**b**) Neighbor–joining tree of the relationships between the five cattle breeds (50 animals), (**c**) Admixture analysis results for five cattle breeds at K = 2 to 4.

**Figure 2 animals-13-03269-f002:**
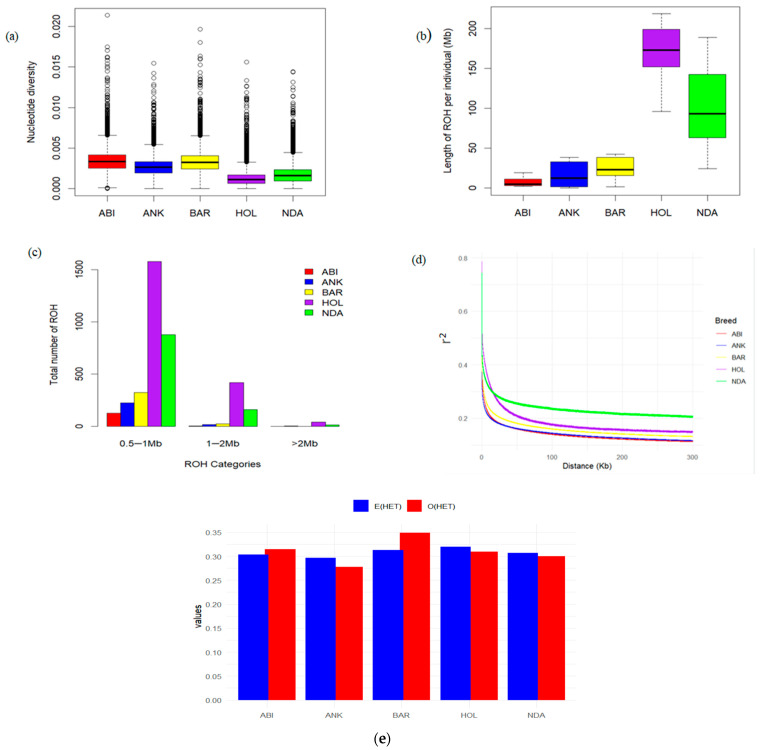
Summary statistics for patterns of genomic variation. (**a**) Average genome-wide nucleotide diversity (100 kb window with 50 kb step size); (**b**) The length of the ROHs in the five studied breeds; (**c**) The distribution of the total number of ROHs in each breed. The median value of this diversity is indicated by a horizontal line within the box, while the box itself represents the first and third quartiles of the distribution. Data points that fall outside the whiskers are considered outliers. (**d**) The genome-wide linkage disequilibrium (LD) decay for each breed; (**e**) The observed (O(HET)) and Expected (E(HET)) heterozygosity of each breed.

**Figure 3 animals-13-03269-f003:**
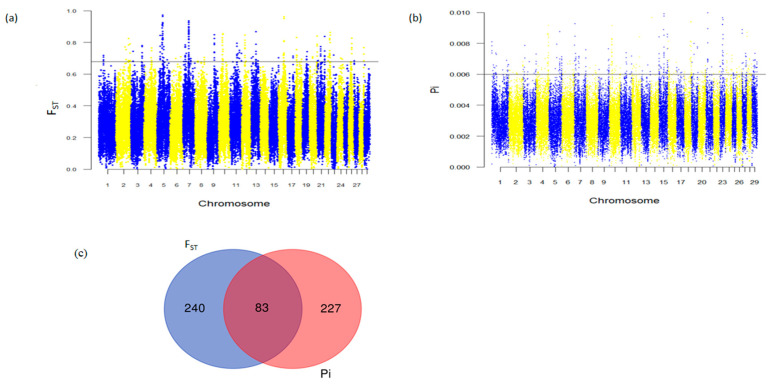
Analysis of selective sweeps in Abigar cattle (**a**) Manhattan plots *F_ST_* selection scan; (**b**) Manhattan plots Pi selection scan; The horizontal dash lines represent the 0.5% outlier regions in both of the selection scan methods; (**c**) Venn diagrams of genes shared by Pi and *F_ST_* selection scan methods.

**Figure 4 animals-13-03269-f004:**
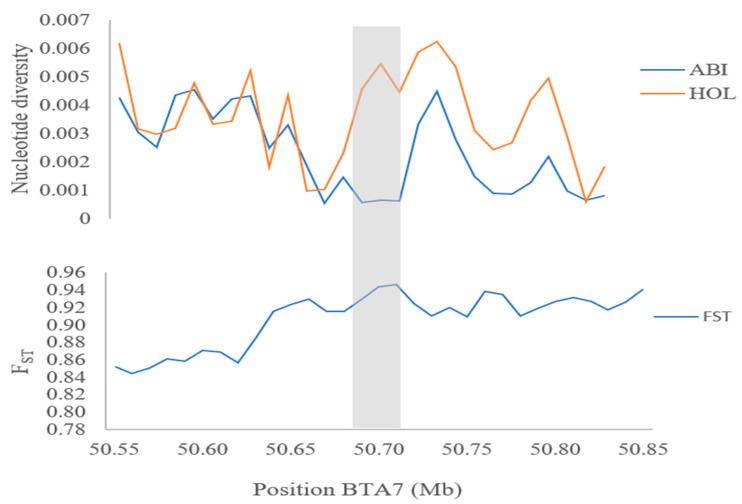
Nucleotide diversity and population differentiation (*F_ST_*, Abigar, and Holstein cattle breeds) plot of DNAJC18 gene.

**Table 1 animals-13-03269-t001:** Genetic differentiation (*F_ST_*) between studied cattle breeds.

	Abigar	Ankole	Barca	Holstein	NDama
Abigar					
Ankole	0.055				
Barca	0.062	0.122			
Holstein	0.275	0.250	0.349		
NDama	0.189	0.173	0.267	0.265	

**Table 2 animals-13-03269-t002:** Candidate genes putatively selected for tropical environment adaptations of Abigar cattle using two selection scan methods (Pi and *F_ST_*).

Methods	BTA	Gene	Ensemble ID	Summary of Gene Function	References
Pi	13	ASIP	ENSBTAG00000034077	Heat tolerance/coat color	[29]
	18	HSF4	ENSBTAG0000000354	Heat tolerance	[30]
	28	SIRT1	ENSBTAG00000014023	Oxidative stress response	[31,32]
*F_ST_*	5	GRIP1	ENSBTAG00000033726	Immune response	[31]
	7	HSPA9	ENSBTAG00000011419	Immune response	[9,33]
Both (Pi and *F_ST_*)	5	IRAK3	ENSBTAG00000007636	Immune response	[8]
	5	HOXC13	ENSBTAG00000000923	Heat tolerance	[6,32,34]
	5	WIF1	ENSBTAG00000014758	Immune response	[9,33]
	7	MZB1	ENSBTAG00000038337	Immune response	[9]
	7	STING1	ENSBTAG00000002296	Immune response	[9]
	7	DNAJC18	ENSBTAG00000002286	Heat tolerance	[9,35]
	7	SLC23A1	ENSBTAG00000010798	Oxidative stress response	[6]
	12	RXFP2	ENSBTAG00000015132	Heat tolerance, horn development	[31]
	16	FAAP20	ENSBTAG00000014136	Dryland adaptation (scarce feed and water supply)	[8]
	16	SKI	ENSBTAG00000038716	Dryland adaptation (scarce feed and water supply)	[8]

## Data Availability

Sequence data will be deposited in GenBank.

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
