# Peer review of "Whole-Genome Resequencing Reveals Selection Signatures of Abigar Cattle for Local Adaptation"

_animals, 2023, doi:10.3390/ani13203269_

Round 1

Reviewer 1 Report

In the present manuscript titled “Whole-genome resequencing reveals Selection Signatures of Abigar Cattle for local Adaptation”, the authors have identified genomic regions and candidate genes responsible for the tropical environmental adaptations of Abigar cattle, which thrive in hostile environmental conditions on the African continent. However, there are some concerns that should be addressed to improve the overall quality of the manuscript.

General questions and comments

The hot and humid environment is characterized by trypanosomiasis. Have the authors identified any candidate genes linked to trypanosomiasis resistance in Abigar cattle?

All references in the manuscript need to be corrected to comply with the journal's formatting guidelines.

I strongly recommend a thorough grammar and language review by a native speaker or a professional grammar agency for the entire manuscript to enhance its readability and clarity.

Page 1, lines 2 remove “and” in the authors list.

Page 1, lines 38-40: please revise the sentence.

Page 3, lines 139-140: Why did you choose for a 100kb window and 50 steps? Have you explored the effects of a smaller window size?

Page 4, line 124: Why did you choose to utilize VQSR instead of hard filtering for variant discovery in downstream analysis? Are there notable advantages to using VQSR over hard filtering?

Page 4 the sentence on lines 148 to 150 appears ambiguous. Please rephrase it for clarity.

Page 4, line 159: Which dataset did you use to construct the NJ tree? Is it based on pruned data or raw data?

Page 5, line 222: Is there evidence of gene flow between Abigar and Ankole cattle, particularly given their proximity in adjacent areas?

Page 8, line 274: Please consider using alternative colors for the Manhattan plots to ensure accessibility for color-blind readers.

Page 8, lines 280: Have you identified any significant KEGG terms related to your primary focus, specifically the adaptation to the local environment?

Page 11, line 374: You mentioned the HOXC12 gene in the text, but there is no corresponding information about this gene in Table 2. Please either include its complete information in the table or provide a citation to the supplementary information.

Page 12, line 440:  Could you clarify what you mean by 'changing circumstances' in relation to your findings? Please specify the context for a clearer understanding.

I recommend the acceptance of the manuscript titled 'Whole-genome resequencing reveals Selection Signatures of Abigar Cattle for local Adaptation'. Despite the minor concerns raised during the review process, the paper offers valuable insights into the genomic adaptations of Abigar cattle to tropical environments, shedding light on the genetic mechanisms underlying their resilience in hostile conditions. The methodology is sound, and the findings contribute to our understanding of local adaptation in livestock. Addressing the suggested revisions and clarifications will enhance the overall quality and readability of the paper, making it a valuable addition to the scientific literature.

Minor editing of English language required.

Author Response

Point by point authors' response

Reviewer 1 The hot and humid environment is characterized by trypanosomiasis. Have the authors identified any candidate genes linked to trypanosomiasis resistance in Abigar cattle? Authors’ response: Although the area is characterized by trypanosomiasis, we did not identify any trypanosomiasis resistance gene in our selection scan analysis. All references in the manuscript need to be corrected to comply with the journal's formatting guidelines. Authors’ response: corrected as suggested I strongly recommend a thorough grammar and language review by a native speaker or a professional grammar agency for the entire manuscript to enhance its readability and clarity. Authors’ response: corrected as suggested Page 1, lines 2 remove “and” in the authors list. Authors’ response: corrected as suggested Page 1, lines 38-40: please revise the sentence. Authors’ response: corrected as suggested Page 3, lines 139-140: Why did you choose for a 100kb window and 50 steps? Have you explored the effects of a smaller window size? Authors’ response: Yes, we tried smaller window size but there is no difference in the identified variants list between small and big windows and step sizes. The only difference is the short window size takes more running time than the large window size. Page 4, line 124: Why did you choose to utilize VQSR instead of hard filtering for variant discovery in downstream analysis? Are there notable advantages to using VQSR over hard filtering? Authors’ response: Compared to traditional hard filtering methods, VQSR offers several advantages, including improved accuracy, adaptability, and comprehensive assessment, reduced false positives, and enhanced validity of our variant calls. Page 4 the sentence on lines 148 to 150 appears ambiguous. Please rephrase it for clarity. Authors’ response: corrected as suggested Page 4, line 159: Which dataset did you use to construct the NJ tree? Is it based on pruned data or raw data? Author’s response: The NJ tree was constructed based on pruned data. Page 5, line 222: Is there evidence of gene flow between Abigar and Ankole cattle, particularly given their proximity to adjacent areas? Author’s response: There is no known gene flow between Ankole and Abigar cattle breeds due to their distant geographical location. Page 8, line 274: Please consider using alternative colors for the Manhattan plots to ensure accessibility for color-blind readers. Authors’ response: corrected as suggested Page 8, lines 280: Have you identified any significant KEGG terms related to your primary focus, specifically the adaptation to the local environment? Author’s response: We did not find significant KEGG relevant to our key findings. Page 11, line 374: You mentioned the HOXC12 gene in the text, but there is no corresponding information about this gene in Table 2. Please either include its complete information in the table or provide a citation to the supplementary information. Authors’ response: Supplementary Table 3 to 5 was cited on page 10 for the HOXC12 gene. Page 12, line 440: Could you clarify what you mean by 'changing circumstances' in relation to your findings? Please specify the context for a clearer understanding. Authors’ response: Corrected as suggested

Reviewer 2 Report

Review reports

Brief summary 

The present study aimed to investigate by whole-genome sequencing:  genomic diversity, population structure and selection signatures of Abigar cattle, an indigenous cattle breed that have developed disease resistance, heat tolerance, and adaptability to harsh environment.

The authors pointed out that this breed exhibit high nucleotide diversity and heterozygosity, with low runs of homozy gosity and linkage disequilibrium, suggesting a genetic landscape less constrained by inbreeding and enriched by diversity

Consequently, this study emphasizes  their potential value for future breeding programs and sustainable livestock management.

General concept comments 

The manuscript is clear, relevant for the field and presented in a well-structured manner. Estimates of genetic diversity represent a valuable resource for biodiversity assessments and are increasingly used to guide conservation and management programs. The experimental design is appropriate and methods section is well detailed as well as statistical analysis. The only point to be made is that a larger number of samples would be recommended for greater accuracy of results.

Conclusions  are consistent and well structured. I suggest in general to include some more recent bibliographical references.

Specific comments 

I suggest to make the bibliography homogeneous.

Author Response

Reviewer 2

General concept comments

The only point to be made is that a larger number of samples would be recommended for greater accuracy of results.

Authors’ response:  Yes, while larger sample sizes are ideal for accuracy, budget constraints often limit us to 10 animals per breed. To address this, most researchers focus on improving genome coverage to enhance result reliability rather than sample size and we did that in our study.

Conclusions are consistent and well-structured. I suggest in general including some more recent bibliographical references.

Authors’ response:  We try to find relevant literature and include it as suggested.   

Specific comments

I suggest making the bibliography homogeneous.

Authors’ response:  Corrected as suggested.

Reviewer 3 Report

Dear authors, Your article is beneficial to scientific knowledge. It is well prepared, with appropriate procedures, material, and discussion. I have a few questions:

I ask the authors to justify that the number of animals analysed is sufficient or to write more about how these individuals were chosen.

 Tables have to be self-explanatory. Therefore, I suggest using the whole name of the breed in tables.

 Why is the analysis more comprehensive for the Abigar breed than for the Barka?

Author Response

Reviewer 3

 I ask the authors to justify that the number of animal analyses is sufficient or to write more about how these individuals were chosen.

Authors’ response: In whole genome sequencing 10 samples are more common for selection signature studies. The number of samples is not the main issue that determines the accuracy of selection signature results rather the depth of the sequence is the main determining factor. Therefore, in this study, we used to sequence with appropriate depth of coverage recommended by different authors. 

 Tables have to be self-explanatory. Therefore, I suggest using the whole name of the breed in tables.

Authors’ response:  Corrected as suggested.

 Why is the analysis more comprehensive for the Abigar breed than for the Barka?

Authors’ response:  We appreciate the reviewer's inquiry regarding the scope of our analysis. It's important to note that our study serves a specific research objective. While we have conducted a comprehensive study involving Barka cattle in a separate context, the focus of this particular study was to perform a genetic comparison between Abigar cattle and indigenous Ethiopian zebu breeds. In the case of Abigar cattle, our aim was to investigate their genetic characteristics, assess potential selective sweeps, and gain insights into their unique genetic makeup in comparison to the local zebu populations. This study was designed to address specific questions related to the Abigar breed, which required a more targeted analysis approach.